# Red Blood Cell Aging as a Homeostatic Response to Exercise-Induced Stress

**Joames K. Freitas Leal** [1] , **Dan Lazari** [1], **Coen C.W.G. Bongers** [2], **Maria T.E. Hopman** [2], **Roland Brock** [1] **and Giel J.C.G.M. Bosman** [1,*]

[1] Department of Biochemistry, Radboud University Medical Center, 6525 GA Nijmegen, The Netherlands; joames.leal@gmail.com (J.K.F.L.); Dan.Lazari@radboudumc.nl (D.L.); Roland.Brock@radboudumc.nl (R.B.)

[2] Department of Physiology, Radboud Institute for Molecular Life Sciences and Radboud Institute for Health Sciences, 6525 GA Nijmegen, The Netherlands; Coen.Bongers@radboudumc.nl (C.C.W.G.B.); Maria.Hopman@radboudumc.nl (M.T.E.H.)

\* Correspondence: Giel.Bosman@radboudumc.nl

**Abstract:** Our knowledge on the molecular mechanisms of red blood cell aging is mostly derived from in vitro studies. The Four Days Marches of Nijmegen in the Netherlands, the world's largest yearly walking event, constitutes a unique possibility to study the effect of mechanical and biochemical stressors occurring during moderate-intensity exercise on red blood cell aging in vivo. Therefore, longitudinal measurements were performed of biophysical, immunological, and functional red blood cell characteristics that are known to change during aging. Our data show that moderate-intensity exercise induces the generation of a functionally improved red blood cell population with a higher deformability and a decreased tendency to aggregate. This is likely to be associated with an early removal of the oldest red blood cells from the circulation, as deduced from the (dis)appearance of removal signals. Thus, the physiological red blood cell aging process maintains homeostasis in times of moderate-intensity exercise-induced stress, probably by accelerated aging and subsequent removal of the oldest, most vulnerable red blood cells.

**Keywords:** aggregation; aging; deformability; red blood cell; exercise

## 1. Introduction

Red blood cell (RBC) homeostasis is maintained through removal of aged, functionally compromised RBCs by macrophages, in combination with erythropoietin-regulated production of young RBCs in the bone marrow. Aging of RBCs is associated with the removal of damaged membrane proteins by the shedding of vesicles, loss of deformability, and oxygen binding, and with the appearance of molecules and/or epitopes that promote recognition and removal by the immune system. The mechanisms underlying functional as well as immunological RBC aging in vivo are likely to be triggered by oxidation, which induces an increase in susceptibility to physiological stress [1,2]. Experimental support for the latter hypothesis is mostly provided by the response of RBCs from healthy donors aged in vivo and RBCs from blood bank concentrates aged in vitro to various stress factors in vitro [1,3]. In addition, recent findings indicate that RBC homeostasis is affected by systemic conditions, such as inflammation, and that altered RBC function and survival affect organismal homeostasis [4–7].

Data on RBC-related parameters obtained during exercise support this concept. In trained athletes, exercise has been reported to induce hemolysis [8,9]. Recently, this has been postulated to be associated with a shift towards a decrease in mean cell age of the RBC population and a concomitant improvement in function [10]. The Four Days Marches of Nijmegen in the Netherlands, the world's largest yearly

voluntary walking event, presents the possibility to study the effect of prolonged, moderate-intensity exercise on the molecular mechanism of RBC aging in vivo. Recent observations in Four Days Marches participants on iron metabolism and cytokine responses have provided indirect indications that changes in the RBC population are involved in the physiological adaptation to prolonged exercise [11–14]. We postulate that these changes consist of an accelerated disappearance of the oldest RBCs, resulting in a functionally improved RBC population. Here, we describe the measurement of a panel of cellular, immunological, and functional parameters of RBC aging to test this hypothesis.

## 2. Materials and Methods

### 2.1. Study Population

Blood was collected by venipuncture using EDTA as anticoagulant from 16 healthy volunteers (11 male, 66 ± 4 years, and five female, 65 ± 6 years) who participated in the 2018 edition of the Nijmegen Four Days Marches (http://www.4daagse.nl/en/), and who were included using the criteria described before [11]. The participants walked 30 kilometers (N = 8), 40 kilometers (N = 7), or 50 kilometers (N = 1) on four consecutive days. All participants gave written informed consent. The study was approved by the local Medical Ethical Committee (CMO Arnhem-Nijmegen; 2007-148) and was in accordance with the Declaration of Helsinki. Blood was collected at baseline (day 0), after day 1, in the morning of day 2, and in the afternoon of day 2. Complete analyses were performed on ten participants who donated blood every day.

Plasma, leukocytes and platelets were removed using Ficoll-Paque [15]. Hemolysis was estimated by measuring the absorbance of the cell-free plasma at 415 nm.

### 2.2. Red blood Cell Fractionation

RBCs were fractionated according to cell density using discontinuous Percoll gradients ranging from 40% Percoll (1.060 g/mL) to 80% Percoll (1.096 g/mL) as described before [15,16]. The various RBC fractions were isolated and washed three times with Ringer's solution [15] by repeated centrifugation for 5 min at 400 g before analysis.

### 2.3. Deformability, Osmotic Fragility, and Aggregation

Deformability and aggregation were measured using a laser-assisted optical rotational cell analyzer (Lorrca MaxSis, Mechatronics, The Zwaag, The Netherlands) as described previously [17].

### 2.4. Flow Cytometry

The percentage of phosphatidylserine (PS)-exposing RBCs was determined using Annexin V, essentially as described before [18]. One million RBCs were incubated with phycoerythrin (PE)-labeled Annexin V (1:25, BD Pharmingen, Hoeven, the Netherlands) for 20 min at room temperature in the dark. Tert-butylhydroperoxide-treated RBCs (1 mM, 45 min at room temperature) served as positive controls. APC-labeled anti-C3d (1 µg/million cells, Assay Pro, St. Louis, Missouri, USA) was used to evaluate the degree of complement-induced opsonization [19]. APC-labeled CD71 (clone CY1G4, 1:200, Biolegend, San Diego, California, USA) was used to measure the percentage of reticulocytes [20]. Staining of band 3 with eosin-5' maleimide (EMA, Thermo Fisher Scientific, Landsmeer, the Netherlands) was performed by incubating one million RBCs with 25 µL of EMA (0.5 mg/mL in Ringer's solution) in the dark at RT for 15 min [21,22].

After staining, RBCs were washed three times with Ringer's solution and analyzed by flow cytometry (FACSCalibur instrument, BD Biosciences, Franklin Lakes NJ, USA) using the CELLQuest software (BD Biosciences). Data were analyzed with the FlowJo cell analysis software v.10 (FlowJo, LLC, Ashland OR) using 200,000 events.

*2.5. Isolation and Characterization of Microvesicles*

Microparticles (MPs) were isolated from the platelet-rich plasma (PRP) obtained after differential centrifugation as described before [4,23]. Microvesicle analysis was performed using mixtures of FITC-labeled CD235a (1:100) and PE/Cy5-labeled CD41 (1:10) by flow cytometry as previously described [4,23]. Sulfate latex microspheres (0.9 μm; Invitrogen, Carlsbad CA, USA) and washed Flow-Count calibration beads (Beckman Coulter, Brea CA, USA) were used for quantification [4,23].

*2.6. Statistical Analysis*

Differences between donors were determined using paired, repeated measures ANOVA followed by Bonferroni correction for multiple comparisons. Two-sided *p*-values less than 0.05 were considered statistically significant.

## 3. Results

Participation in the Four Days Marches led to an overall steady increase in RBC volume, as indicated by the increase in the mean value of the forward scatter, and to a concomitant decrease in the mean fluorescence intensity of the membrane protein probe EMA (Figure 1).

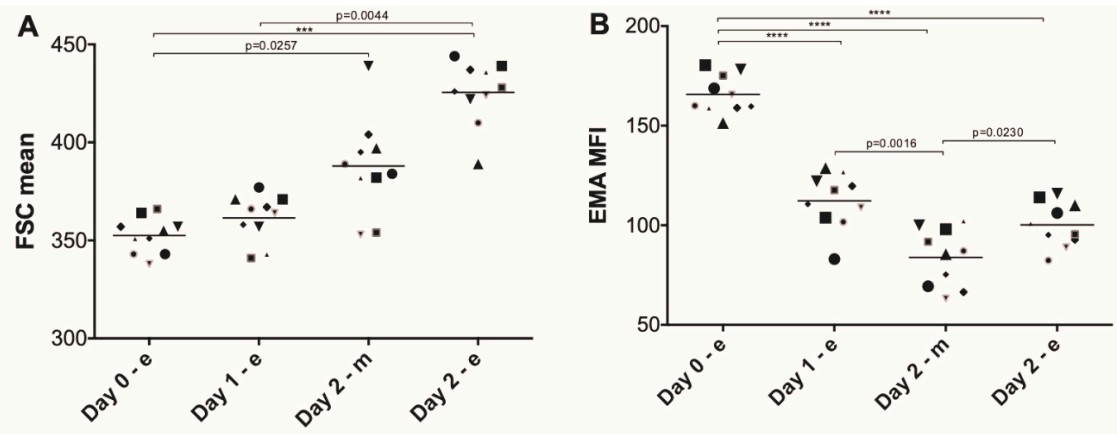

**Figure 1.** Changes in forward scatter and eosin-5′ maleimide (EMA) binding of red blood cells during the Nijmegen Four Days Marches. (**A**) Mean forward scatter (FSC) extracted from the flow cytometry data; (**B**) the mean fluorescent intensity (MFI) after binding of EMA, as described in Materials and Methods. Each symbol represents one donor. Significant differences: ***, *p* < 0.001; ****, *p* < 0.0001 (N = 10).

These changes in cell volume are accompanied by changes in cell density, as indicated by qualitative analysis of the RBC distribution over Percoll gradients (Figure S1).

In all subjects, hemolysis slightly increased during the marching days from 0.25 ± 0.03 to 0.39 ± 0.14 absorption units (N = 14). This increase in intravascular hemolysis was accompanied by a decrease in the fraction of phosphatidylserine-exposing RBCs after each day of marching (Figure 2A). Moreover, the fraction of C3d-positive RBCs increased significantly in the morning before the second day of marching (Figure 2B, day 2-m).

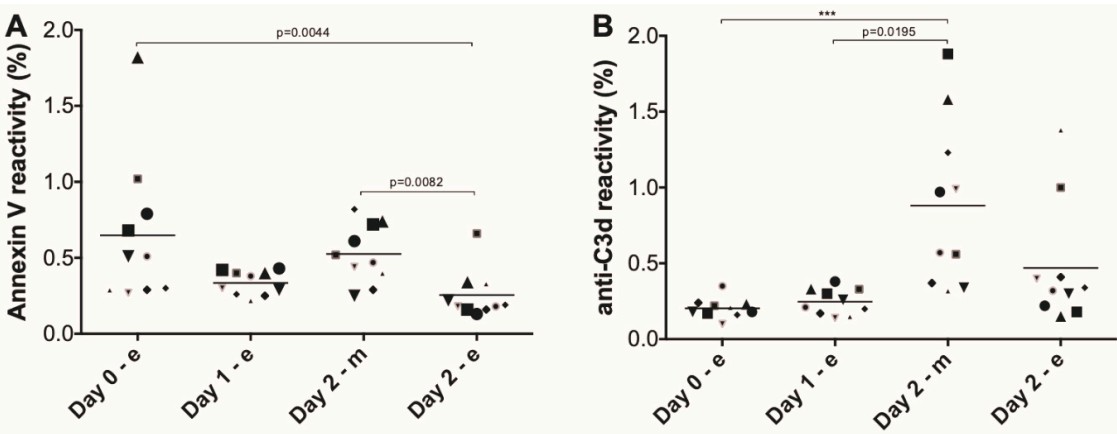

**Figure 2.** Appearance of removal signals on red blood cells during the Nijmegen Four Days Marches. (**A**) The fraction of phosphatidylserine (PS)-exposing red blood cells (RBCs) as determined using fluorescent annexin V; (**B**) the fraction of C3d-positive RBCs, as determined with a fluorescent monoclonal antibody, as described in Materials and Methods. Each symbol represents an individual. Significant differences: ***, $p < 0.001$; **** (N = 10).

The fraction of the most mature reticulocytes, as detected by the fraction of CD71-positive RBCs [20], decreased after the first day of marching, but increased again during the consecutive night (Figure 3).

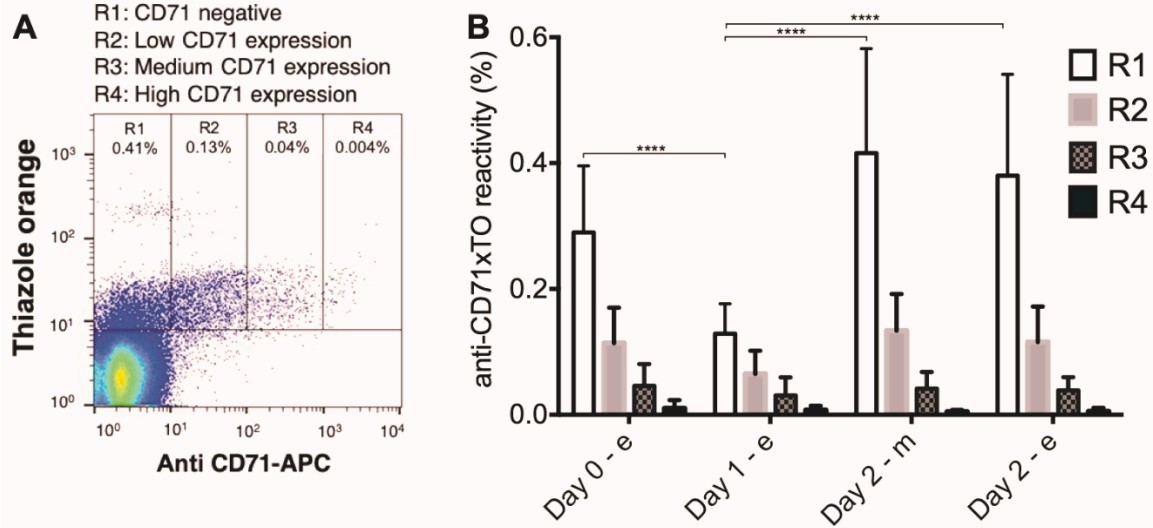

**Figure 3.** Changes in the fraction of reticulocytes during the Nijmegen Four Days Marches. (**A**) The numbers of reticulocytes were determined using fluorescent anti-CD71 antibody in combination with thiazole orange: R1, CD71 negative ($p < 0.001$), R2, low CD71 expression ($p = 0.0362$), R3, medium CD71 expression ($p > 0.05$), R4 high CD71 expression ($p > 0.05$); (**B**) percentage of each reticulocyte maturation fraction group per day. ***, $p < 0.001$; ****, $p < 0.0001$ (N = 10).

RBC aging is accompanied by the generation of microvesicles [24]. In the blood of the Four Days Marches participants, we found no significant changes in RBC-derived microvesicle concentrations. The concentrations of platelet-derived microvesicles, however, increased significantly after each marching day (Figure 4).

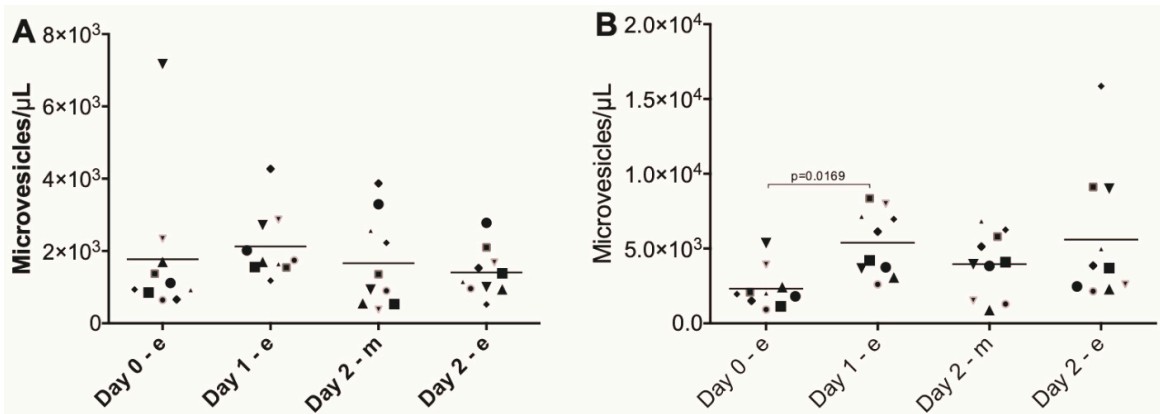

**Figure 4.** RBC-derived and platelet-derived microvesicles in the blood of participants in the Nijmegen Four Days Marches. (**A**) Concentration of RBC-derived microvesicles (ANOVA main effect $p > 0.05$); (**B**) concentration of platelet-derived microvesicles (ANOVA main effect $p = 0.0169$). Microvesicles were isolated from the blood of Four Days Marches participants and their concentrations were measured using RBC-specific and platelet-specific markers as described in Materials and Methods (N = 10).

Aggregation and deformability are key characteristics of RBC function [25]. Participation in the Four Days Marches induced a decrease in RBC aggregation after the first day, and an increase in deformability after two days (Figure 5).

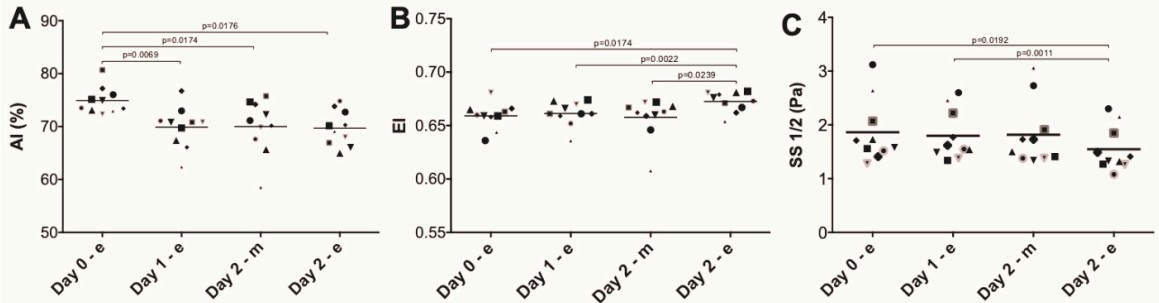

**Figure 5.** Red blood cell deformability and aggregation during the Nijmegen Four Days Marches. The tendency to aggregate (Aggregation Index, AI in panel (**A**); ANOVA main effect $p = 0.0043$), maximal deformability (Elongation Index, EI in panel (**B**); ANOVA main effect $p = 0.0027$), and the elongation kinetics during increased stress (SS $\frac{1}{2}$ in panel (**C**); ANOVA main effect $p = 0.0022$) were measured using an LORRCA ektacytometer as described in Materials and Methods. Post-test significant differences (N = 10).

## 4. Discussion

Participation in the Nijmegen Four Days Marches constitutes a moderate-intensity exercise, of approximately 70 percent of the maximal heart rate [11]. This provides an excellent opportunity to investigate the molecular mechanisms involved in the aging process of human RBCs in vivo. Our present data from a relatively small but representative number of subjects [1,13,14,18] indicate that participation in this event induces an early homeostatic response in the participants' RBCs. We postulate that this response results in a shift towards an overall younger RBC population, caused by accelerated aging and disappearance of the oldest cells from the circulation. These processes may be responsible for the changes in hemoglobin content starting on the first marching day [13]. The main findings that support this hypothesis are an increase in cell volume (Figure 1) with an accompanying shift in cell density (Supplementary Material Figure S1). These changes were associated with improved function, as indicated by an increase in deformability and a decrease in the tendency to aggregate (Figure 5). Similar alterations with a concomitant increase in function, indicating a shift towards an overall younger RBC population, have been reported to result from various forms of exercise [10,26–29].

Preferential disappearance of the oldest RBCs might explain the decrease in tendency to aggregate, since increased aggregation is associated with an aging-associated decrease in cell surface charge [30,31]. The appearance of phosphatidylserine in the outer leaflet of the RBC membrane and the accumulation of activated complement (Figure 3) contributes to an increase in phagocytosis. This affects especially the oldest RBCs in the population, since they are the most susceptible to hyperosmotic conditions in vitro [1,3,17]. The same stress mechanism may be operative in vivo, in view of the light hypernatremia due to inadequate fluid intake in the first days of the Four Days Marches observed before [11]. Exercise is accompanied by increased oxidation and methemoglobin formation, especially in the oldest RBCs [2,32]. It is possible that this increased susceptibility to oxidation is responsible for the small increase in hemolysis. This is in line with the decrease in haptoglobin and increase in ferritin after the first march day [13].

In vitro, physiological stressors induce RBC vesiculation [4,5], but we did not observe increased RBC-derived microvesicle concentrations in the blood of our participants (Figure 5). This may be due to the fast removal of RBC-derived microvesicles by the reticulo-endothelial system [33]. However, we observed a significant increase in the concentration of platelet-derived vesicles (Figure 4), supporting the suggestions that exercise may activate platelets and induce platelet precursor mobilization [34–37]. Platelet activation may be increased by exercise-induced inflammatory responses [8,14]. Similarly, inflammation has a pronounced effect on RBC membrane organization and stability [4,5,38].

Mechanical stimulation plays a role in the final differentiation of cultured RBCs in vitro [20]. Thus, an exercise-associated increase in blood flow may stimulate exosome-mediated removal of RNA and fragments of cellular organelles, and may thereby accelerate the maturation of reticulocytes in vivo (Figure 4).

In conclusion, our data support the hypothesis that exercise causes various forms of physiological stress such as oxidation, increased shear stress, and inflammation [2,8,13,14] that induce an acceleration of the removal of the oldest, most susceptible RBCs. This occurs already during the first day of prolonged moderate-level exercise and may represent an early homeostatic adaptation leading to a functionally improved RBC population. Prolonged and/or intensive exercise results in increased RBC aging and removal, which induces later adaptations, such as erythropoietin-stimulated RBC production [13,14]. Our data provide a molecular framework for these changes and support current theories on the cellular mechanisms of physiological aging and removal of RBCs in vivo, which so far have been mostly deduced from observations and interventions in vitro.

**Supplementary Materials:** The following are available online at http://www.mdpi.com/2076-3417/9/22/4827/s1, Figure S1: Percoll gradients of red blood cells during the first two days of the 2018 edition of the Nijmegen Four Days Marches. Blood was taken at baseline (day 0-e), in the evening of day 2 (day 1-e), in the morning before day 2 (day 2-m), and after day 2 (day 2-e). After removal of plasma, platelets, and white blood cells, RBCs were separated according to density using Percoll gradients, as described in Materials and Methods. The figure shows the representative results of one donor: A shift towards denser RBCs after the first day of marching (arrows) and a shift towards lighter RBCs during the second day (arrow head). Similar patterns were seen in all donors.

**Author Contributions:** M.T.E.H., R.B., and G.J.C.G.M.B. were involved in designing the study; J.K.F.L., D.L., and C.C.W.G.B. collected the samples and performed the measurements and first analyses; J.K.F.L. and G.J.C.G.M.B. wrote the first draft of the manuscript; all authors contributed to the final manuscript.

**Funding:** The research of JKFL was supported by the National Council for Scientific and Technological Development—CNPq—Brazil.

**Acknowledgments:** We thank all the people involved in the Four Days Marching study.

**Conflicts of Interest:** The authors declare no conflict of interest.

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
