# Peer review of "Red Blood Cell Aging as a Homeostatic Response to Exercise-Induced Stress"

_applsci, doi:10.3390/app9224827_

Round 1

Reviewer 1 Report

In this study Leal and colleagues take an opportune event of defined moderate exercise event to examine the effect on RBC aging and the possible dynamics involved in cellular kinetics-clearance and production in vivo. It is an elegant study well written and presented. I believe the data is well interrogated, interpreted and concisely presented in a very clear fashion. I believe the paper will be of interest to a wide range of readers- from sports scientists, clinical exercise medicine and physiologists. It is an informative insight into RBC homeostasis.

Author Response

We thank the reviewer for these comments.

Reviewer 2 Report

The authors described a study of rad blood cell aging in response to the Four Days Marches. I have some concerns:

The results are based on 10 subjects, which seems to be limited. The limitations should be discussed. The analysis did not include any covariates such as age or sex, which could potentially bias the result. This is a 4-day event, but the blood samples were only collected on the first 2 days. An extension of the sample collection to days 3 and 4 would be helpful in determining the long-term trend. The authors used repeated measures ANOVA or Friedman test. However, it is unclear which test was used specifically in the Result section.

Author Response

The authors described a study of rad blood cell aging in response to the Four Days Marches. I have some concerns:

The results are based on 10 subjects, which seems to be limited. The limitations should be discussed.

Response: Based on previous data from the Four-Day Marches studies and the known intra- and interindividual variability in RBC aging parameters (refs. 1, 13, 14, 18), in combination with the selection criteria (ref 11) makes these subjects a representative group. These considerations have been added to the Discussion (line 155).

The analysis did not include any covariates such as age or sex, which could potentially bias the result. This is a 4-day event, but the blood samples were only collected on the first 2 days. An extension of the sample collection to days 3 and 4 would be helpful in determining the long-term trend.

Response: We agree with the reviewer that an extension might have been helpful to estimate long-term adaptations. This was, unfortunately, hampered by logistical (and psychological) circumstances. Also, based on the literature data and on previous, comparable data on 4-day participants, it is likely that long-term adaptations require more time (e.g. ref 13). This issue/consideration and relevant reference have been added to the Discussion (line 191-193).

The authors used repeated measures ANOVA or Friedman test. However, it is unclear which test was used specifically in the Result section.

Response: All results could be analyzed by ANOVA/Bonferroni for parametric distribution. This has been adjusted in the Materials and Methods section.

Reviewer 3 Report

The manuscript by Joames K. Freitas Leal et al. reports that moderate-intensity exercise-induced stress impact to physiological red blood cell aging process probably by accelerated aging and subsequent removal of the oldest, most vulnerable red blood cells. 

In my opinion Four Days Marches of Nijmegen in the Netherlands has been a really good impact to study the treated argument. Moreover, I appreciate the idea of in vivo study and how this study is performed. However, I would improve the discussion to answer to these questions:

lane 187/188 "exercise causes various forms of physiological stress". What type of physiological stress? What are the differences about moderate or intense exercise relating to red blood cells? lane 189/190 "and may represent an early homeostatic adaptation leading to a functionally improved RBC population". How could it reflect to the other organs?s physiology? During moderate exercise, could have I the same answer? And if I have a sedentary life, how is the red blood behavior?

In conclusion, as it is a scientific article with an informative applicative typology, I appreciate to improve the discussion about control case (sedentary or moderate exercise level) to better understand what is the border and about this type of red blood behavior what are the consequences derived.

Thanks a lot 

Author Response

The manuscript by Joames K. Freitas Leal et al. reports that moderate-intensity exercise-induced stress impact to physiological red blood cell aging process probably by accelerated aging and subsequent removal of the oldest, most vulnerable red blood cells. 

In my opinion Four Days Marches of Nijmegen in the Netherlands has been a really good impact to study the treated argument. Moreover, I appreciate the idea of in vivo study and how this study is performed. However, I would improve the discussion to answer to these questions:

lane 187/188 "exercise causes various forms of physiological stress". What type of physiological stress?

Response: Oxidation, shear stress, inflammation seem to be the most relevant stressors. This has been added to the text (line 189).

What are the differences about moderate or intense exercise relating to red blood cells?

Response: This issue is now addressed (lane 191-193).

lane 189/190 "and may represent an early homeostatic adaptation leading to a functionally improved RBC population". How could it reflect to the other organs?s physiology? During moderate exercise, could have I the same answer? And if I have a sedentary life, how is the red blood behavior?  

Response: A short discussion of these issues has been added to the Discussion (lane 191-192)

In conclusion, as it is a scientific article with an informative applicative typology, I appreciate to improve the discussion about control case (sedentary or moderate exercise level) to better understand what is the border and about this type of red blood behavior what are the consequences derived.

Round 2

Reviewer 2 Report

This is a revision and the authors have addressed my previous concerns.

Author Response

Reviewer: the authors have addressed my previous concerns

Response: We thank the reviewer for the constructive criticism

Reviewer 3 Report

Thanks for the answers and your improvement.

I accept in the present form

Author Response

Reviewer: I accept in present form

Response: we thank the reviewer for the constructive criticism